# Parent-Reported Perceived Cognitive Functioning Identifies Cognitive Problems in Children Who Survived Neonatal Critical Illness

**DOI:** 10.3390/children9060900

**Published:** 2022-06-16

**Authors:** Yerel Ilik, Hanneke IJsselstijn, Saskia J. Gischler, Annabel van Gils-Frijters, Johannes M. Schnater, Andre B. Rietman

**Affiliations:** 1Department of Pediatric Surgery and Intensive Care, Erasmus MC Sophia Children’s Hospital, Dr. Molewaterplein 40, 3015 GD Rotterdam, The Netherlands; 404898yi@eur.nl (Y.I.); h.ijsselstijn@erasmusmc.nl (H.I.); s.gischler@erasmusmc.nl (S.J.G.); a.vangils-frijters@erasmusmc.nl (A.v.G.-F.); j.schnater@erasmusmc.nl (J.M.S.); 2Department of Child and Adolescent Psychiatry/Psychology, Erasmus MC Sophia Children’s Hospital, Dr. Molewaterplein 40, 3015 GD Rotterdam, The Netherlands

**Keywords:** pediatric perceived cognitive functioning, PedsPCF, neuropsychological assessment, executive functioning, behavior rating inventory of executive function, BRIEF, congenital diaphragmatic hernia, esophageal atresia, neonatal extracorporeal membrane oxygenation

## Abstract

Children with congenital anatomical foregut anomalies and children treated with neonatal extracorporeal membrane oxygenation (ECMO) are at risk for neurocognitive morbidities. We evaluated the association between the parent-reported pediatric perceived cognitive functioning (PedsPCF) questionnaire and the parent-reported behavior rating inventory of executive function (BRIEF) as well as neuropsychological assessments (NPA). We included 8-, 12- and 17-year-old participants who had joined a prospective follow-up program between 2017 and 2019. Self- and parental proxy-reported PedsPCF and proxy-reported BRIEF scores and their mutual association were evaluated. In total, 168 participants were included. Self- and proxy-reported PedsPCF scores were significantly below normal (mean (SD) z-score: −0.35 (0.88), *p* < 0.001; −0.36 (1.06), *p* < 0.001, respectively). Total BRIEF scores were significantly above normal (mean (SD) z-score 0.33 (0.98), *p* < 0.001). Proxy-reported PedsPCF scores and the Metacognition Index subscores of the BRIEF correlated strongly (τ = 0.551, *p* < 0.001). Self-reported PedsPCF scores were not associated with NPA test scores. Proxy-reported PedsPCF scores were positively associated with multiple NPA test scores, especially intelligence (*R*^2^ = 0.141). The proxy-reported PedsPCF revealed cognitive problems more often than the BRIEF in school-aged children who had survived neonatal critical illness. The proxy-reported PedsPCF may support clinical decision-making regarding the need for extensive neuropsychological assessments.

## 1. Introduction

Survival rates for children born with a congenital anatomical anomaly such as a congenital diaphragmatic hernia (CDH) or esophageal atresia (EA) have increased over the years [1]. The introduction of extracorporeal membrane oxygenation (ECMO) has improved survival rates of neonates suffering from severe respiratory failure [2]. This decrease in mortality, however, has resulted in an increase in long-term morbidity. Previous research has shown that these children generally have an intelligence within the range of the reference population, but show memory and attention deficits at school age [3,4,5,6,7,8,9]. These deficits are suggested to be the cause of a common neurodevelopmental pathway causing hippocampal alterations, irrespective of underlying diagnosis, induced by common conditions associated with neonatal critical illness such as hypoxia, neuroinflammation, and exposure to anesthetics [10].

A longitudinal follow-up program for children with congenital anomalies and/or those treated with neonatal ECMO is being run in our tertiary level hospital since 1999. The program includes standardized screening of physical health, motor function, cognition, executive functioning, and quality of life at set ages [9,11,12,13,14]. In previous studies, neuropsychological assessments (NPA) revealed that children born with EA, CDH, and those treated with neonatal ECMO have problems in attention, memory, and executive functioning [3,4,5,6,7,8,9].

Results of NPA at school age can facilitate early recognition of neurodevelopmental deficits and therefore offer the opportunity for timely intervention. However, NPA are labor-intensive and time-consuming, and instruments that are more efficient for assessing neurocognitive development are being sought. Lai and coworkers developed the pediatric perceived cognitive functioning questionnaire (PedsPCF) to screen for neurocognitive deficits in children with cancer [15,16]. This 30-item questionnaire evaluates perceived cognitive functioning by the child (self-rating) or the parent (proxy-rating). Lai and coworkers suggested that research to evaluate the PedsPCF clinical usefulness should focus on the association between the PedsPCF and tests that objectively measure cognitive function [15]. In 2019, a Dutch version of the PedsPCF was validated [17]. This 10-item Dutch version correlated strongly (r = 0.94, *p* = <0.001) with the scores of the 30-item form [17]. Several clinical studies have used the PedsPCF to measure cognitive functioning in pediatric patients. For children born with giant omphalocele, multiple congenital anomalies or gastroschisis, the PedsPCF scores appeared to be lower than those of their healthy matched controls, indicating poorer cognitive functioning [18,19]. The PedsPCF also differentiated between different leukoencephalopathy grades for children with brain tumors [20].

Nevertheless, research focusing on the association between the PedsPCF and NPA is still limited. To our knowledge, only one study described the association between the self-reported and proxy-reported PedsPCF with neuropsychological testing results, and showed almost no significant correlation for children with minimal hepatic encephalopathy [21].

We aimed to evaluate the additional value of the PedsPCF within our follow-up program for children with congenital foregut anatomical anomalies and/or neonatal ECMO treatment by assessing the association between the PedsPCF and the frequently used behavior rating inventory of executive function (BRIEF) [22,23] and the association between the PedsPCF and NPA.

## 2. Materials and Methods

### 2.1. Participants

In this observational cohort study, we included participants of our prospective longitudinal follow-up program who underwent NPA at 8, 12, or 17 years and who themselves and/or their parents filled in the PedsPCF between September 2017 and December 2019. We excluded children who could not be assessed with standardized instruments due to genetic syndromes affecting neurocognitive development or due to severe neurodevelopmental impairment. The local Medical Ethics Review Board waived approval (‘Medical Research in Human Subjects Act does not apply to this research proposal’; MEC-2016-111).

### 2.2. Data Collection

We collected data on diagnosis, gestational age, birth weight, duration of initial ventilation, and hospital stay from electronic medical records. PedsPCF, BRIEF, NPA, and school functioning were all administered at one follow-up moment at 8, 12, or 17 years of age. Parents completed the PedsPCF and the BRIEF questionnaires in the waiting room, whereas the children completed the PedsPCF during their assessment with the psychologist. All children underwent NPA by a licensed developmental psychologist. At 8 years, a full intelligence assessment was completed using the WISC-III-NL [24]. At 12 years of age, intelligence was measured with an abbreviated version of the WISC-III-NL, and at 17 years of age, intelligence was assessed using the WAIS-IV-NL [25,26]. During each assessment, attention, verbal memory, visuospatial memory, visuospatial processing, and executive functioning were also tested, by applying validated neuropsychological tests [24,25,26,27,28,29,30,31,32,33]. During a semi-structured interview, the psychologist collected information on the child’s school performance regarding level, grade, and extra help at school.

### 2.3. Measures

The Dutch version of the PedsPCF consists of 30 items and assesses various neurocognitive domains including memory, attention, and executive functioning. Since previous analyses pointed out that the PedsPCF is unidimensional, there is just a single Peds-PCF score [16,17]. A high raw PedsPCF score represents better perceived cognitive functioning [15]. To compare total scores of the 30-item form with those of the 10-item form, the 10-item scores were extracted from the completed PedsPCF afterwards. The BRIEF consists of 75 questions and provides next to a total score, scores for two subscales, the Behavior Regulation Index (BRI) and the Metacognition Index (MCI). The BRI captures the rated child’s ability to shift cognitive set and modulate emotions and behavior via appropriate inhibitory control. The MI reflects the rated child’s ability to initiate, plan, organize, self-monitor, and sustain working memory. For the BRIEF, higher scores represent worse functioning. To facilitate comparisons, we reversed these scores such that lower scores represent worse functioning [22,23]. Reference population data for the questionnaires and NPA were obtained from large population samples, and if available, from the Dutch population. School functioning was recorded by the psychologist and categorized as: regular—no learning disabilities, no remedial teaching, regular education; regular with help—learning disabilities and/or remedial teaching, regular education; special education—school for children with learning disabilities and/or behavioral disorders. Intelligence, attention, verbal and visuospatial memory, and executive functioning were measured using validated Dutch tests (Appendix A: Overview of neuropsychological assessments, Appendix A: Description of neuropsychological assessments).

### 2.4. Data Analyses

If responses to 6 or more questions from the 30-item PedsPCF were missing, scores were excluded for further analysis. If less than six items were missing at random, the scores for those items were calculated by mean imputation. If responses to 2 or more questions of the extracted 10-item short version were missing, scores were excluded for further analysis. BRIEF and PedsPCF raw scores were first converted into t-scores using Dutch normative values (mean t-score = 50, SD = 10) and then converted into z-scores (mean z-score = 0, SD = 1).

Scores on the intelligence scale are represented by standard scores (population mean M = 100, SD = 15). NPA results were converted into z-scores (mean z-score = 0, SD = 1). If necessary, z-scores were inverted so that a higher score always indicated better performance. To limit the number of extreme negative outliers in Dot Cancellation test and Stroop test scores, these scores were truncated to −3. Z-scores lower than or equal to −1 were regarded as likely to represent impaired cognitive functioning [15,28].

In descriptive statistics, categorical variables are presented as number (%), continuous variables as mean (SD), and non-normally distributed continuous variables as median (IQR).

Total self-reported and proxy-reported PedsPCF scores, total BRIEF scores, BRI, MCI, school functioning, and NPA outcomes were compared with normative data using one-sample *t*-tests and chi-squared tests for comparing means and proportions. To assess the utility of the PedsPCF, we determined the association between PedsPCF and BRIEF and the association between PedsPCF and NPA. To compare the PedsPCF with the BRIEF, total scores and subscale scores were analyzed using Kendall’s tau correlation. Simple linear regressions were performed to analyze the association between the PedsPCF and tests of the standardized NPA that represent subdomains measuring concepts comparable to the concepts underlying the PedsPCF [16].

One-way analysis of variance (ANOVA) or Kruskal–Wallis ANOVA was used to analyze the relation between school functioning and PedsPCF and BRIEF scores. To assess the association of the 10-item PedsPCF and the 30-item PedsPCF, scores were compared using Kendall’s tau correlation. Subgroup analyses were performed using ANOVA or Kruskal–Wallis ANOVA. Corresponding to the statistical methods, effect sizes were calculated according to Cohen’s d, r, and Cohen’s f, which resulted in the following interpretations [34]:Cohen’s d (= mean–population mean/sample SD): small = 0.2, medium = 0.5, large = 0.8.r (= standardized test statistic/√N): small = 0.1, medium = 0.3, large = 0.5.Cohen’s f (= (√η^2^/1 − η^2^)): small = 0.1, medium = 0.25, large = 0.4.Strengths of the Kendall tau correlation coefficients were interpreted as followed [35]: τ: weak = 0.1, moderate = 0.4, strong = 0.7.

Statistical analyses were performed using SPSS V.25.0. All statistical tests were performed two-sided with a significance level of 0.05.

## 3. Results

### 3.1. Participants

In total, 168 children were included, quite equally divided over the assessment ages of 8, 12, and 17 years (Table 1) (*p* = 0.272). Of the included children, 134 self-reports and 159 proxy-reports were analyzed. The prevalence of children in this cohort attending special education was twice as high as compared to the Dutch reference population (*χ^2^* = 7.701, *p* = 0.006) [36,37].

### 3.2. Questionnaire Outcomes

In total, 134 children had sufficient time during the neuropsychological assessment to complete the self-reported PedsPCF, resulting in a response rate of 80%. The proxy-reported PedsPCF and the BRIEF were completed by 160 parents (95%).

Total self- and proxy-reported PedsPCF scores were significantly lower in our study population than in the reference population (both *p* < 0.001; Table 2). Total BRIEF z-scores were significantly higher than the norm data (*p* < 0.001). For the self-reported PedsPCF, the proportion of children with scores lower than or equal to −1 SD was 22%. For the proxy-reported PedsPCF, this was 30%. For the parent-rated BRIEF, only 9% of the scores were lower than or equal to −1 SD. Only two parents reporting low BRIEF scores (13%) reported PedsPCF scores within the normal range. However, of the seven children whose parents reported executive functioning problems (i.e., BRIEF scores below −1 SD), three considered their cognitive functioning as normal (i.e., total score PedsPCF self-report above −1 SD). When looking at the different age groups (children of 8, 12, or 17 years old), no significant differences were found regarding total PedsPCF scores (self-reported: *F* = 0.098, *p* = 0.907; parent-reported: *F* = 0.324, *p* = 0.724).

### 3.3. Associations of PedsPCF with BRIEF, Outcomes of NPA, and School Functioning

Self-reported PedsPCF scores correlated significantly but weakly with all three BRIEF scores (total BRIEF: *τ* = 0.265, *p* < 0.001; BRI: *τ* = 0.205, *p* = 0.001; MCI: *τ* = 0.263, *p* < 0.001). For proxy-reported PedsPCF, correlations were stronger and significant for all three BRIEF scores, resulting in weak to moderate correlations (total BRIEF: *τ* = 0.552, *p* < 0.001; BRI: *τ* = 0.366, *p* < 0.001; MCI: *τ* = 0.551, *p* < 0.001). For the correlations between the proxy-reported PedsPCF and BRIEF scores, we refer to Figure 1.

Overall, intelligence was not significantly different from that of the norm population (*p* = 0.951; Table 3). The outcomes of NPA in different subgroups are shown in Appendix A.

Simple linear regression analyses did not reveal any variable that was significantly associated with the total self-reported PedsPCF score (Appendix A: Simple regressions per variable for self-reported PedsPCF). Table 4 presents the simple linear regressions performed for total proxy-reported PedsPCF scores. Total IQ was most prominently associated with total proxy-reported PedsPCF score with *R*^2^ = 0.141 (*p* < 0.001). For all tested subdomains of cognition, multiple variables appeared to be significantly related with the total proxy-reported PedsPCF score with medium to large effect sizes. The assumptions for performing simple linear regressions were checked beforehand [38].

Figure 2 shows clustered box plots of total self-and proxy-reported PedsPCF scores and total BRIEF scores for the three categories of school functioning. Interestingly, children attending special education reported higher self-reported PedsPCF scores than children in the category ‘regular’ or ‘regular with help’, but Kruskal–Wallis ANOVA showed no statistically significant differences (H = 4.605, *p* = 0.100, Cohen’s f = 0.19). For proxy-reported PedsPCF scores, statistically significant differences between the different school functioning categories were found with a medium effect size (H = 16.991, *p* < 0.001, Cohen’s f = 0.35) with a lower PedsPCF score for children who needed additional help at school or attended special education. Total BRIEF scores also differed significantly for the different school functioning categories but with a small effect size in the same direction (F = 4.082, *p* = 0.019, Cohen’s f = 0.23).

### 3.4. Associations of 30- and 10-Item PedsPCF

For the 30- and 10-item versions of the PedsPCF, Kendall’s tau correlation was significant for self- and proxy-reported scores with a strong association (self-reported: τ = 0.801, *p* = 0.000; proxy-reported: τ = 0.837, *p* < 0.001). The association between the 30-item and 10-item PedsPCF is shown in Appendix A.

## 4. Discussion

We performed this study to assess the additional value of the PedsPCF in the follow-up for children with anatomical foregut anomalies and/or those who were treated with neonatal ECMO. We compared PedsPCF and BRIEF scores with population norm data and examined their mutual association as well as the association between PedsPCF and NPA scores. Our study population scored significantly lower on the self- and proxy-reported PedsPCF compared to normative data. The proxy-rated BRIEF scores, however, were significantly higher than the norm scores. As the mean z-score was 0.33, we regarded this difference not to be clinically relevant. Self-reported PedsPCF scores were all weakly associated with BRIEF scores and not associated with NPA test scores or level of school functioning. Next to this, weak to moderate correlations between proxy-reported PedsPCF and BRIEF scores were found, suggesting their overlap but also their complementarity when mapping the perceived cognitions of these patients. Total proxy-reported PedsPCF scores were significantly associated with multiple NPA test scores. These findings suggest that the proxy-reported PedsPCF in particular can provide a first insight into the child’s neurocognitive performance. Since the 10-item form correlated good with the 30-item form, the short form may already provide sufficient information.

For children born with congenital anatomical malformations, lower scores on the PedsPCF compared to scores of healthy controls have been found in previous studies [18,19]. The moderate correlation found in the present study between the proxy-reported PedsPCF and Metacognition Index score of the proxy-reported BRIEF is similar to that found in a pilot study by Ohnemus and coworkers [21]. This moderate relation could be assigned to the partly overlapping neurocognitive domain investigated in that pilot study, i.e., executive functioning, and the fact that both questionnaires are proxy-reported. On the other hand, the weak correlations between the PedsPCF and the BRIEF’s Behavior Regulation Index subscale could indicate that this particular part of the BRIEF measures additional information that is different from and possibly complementary to the PedsPCF. However, since our study population did not have deficient scores on the BRIEF, it could be argued that the BRIEF is less suitable for detecting cognitive problems in this particular clinical group.

In line with previous research, our study group as a whole had an average IQ within the normal range and showed significantly lower scores on multiple NPA tests [3,4,5,6,7,8,9]. To our current knowledge, only one previous study has explored the associations between the PedsPCF and NPA [21]. In this study, no significant correlation between the PedsPCF and neuropsychological testing for children with minimal hepatic encephalopathy was found, except for a positive association of self-reported PedsPCF scores with one attention test score (r = −0.70) [21]. The discrepancy between this study and the current study may be due to differences in severity of illness and age of onset; generally, children with minimal hepatic encephalopathy were and are less critically ill and are older than children who suffered from neonatal critical illness [10,39]. Another explanation may be that we assessed a broader range of neuropsychological tasks. Nevertheless, we found no association between self-reported PedsPCF and NPA variables, suggesting that, unlike the proxy-reported PedsPCF, the self-reported PedsPCF cannot be used as a screening tool for cognitive problems.

The weak correlation between proxy-reported PedsPCF scores and NPA outcomes found in the present study suggests that some children overestimate themselves and others underestimate themselves with regard to their objectively measured performance. However, the self-reported PedsPCF may still provide additional information about the self-perceived cognitive problems of these children in everyday life. Parents’ perception appears to be more consistent with objectively measured cognitive functions. The absence of associations between pediatric self-reports and clinical outcomes has been described before [14]. We believe that early recognition of neurocognitive problems is important, even when these problems do not seem to affect functioning in everyday life, since school-aged children cannot yet see the potential impact.

With respect to school functioning, scores on the proxy-reported questionnaires appear to be in line with the ordinal levels, i.e., decreasing scores for children in the categories ‘regular with help’ and ‘special’. This finding suggests that the child’s school functioning level is reflected in parent-reported perceived cognitive functioning, which may be regarded as a validation of the proxy-reported PedsPCF. Mean total BRIEF scores across all levels were higher than mean proxy-reported PedsPCF scores, suggesting that the information from the parent-perceived PedsPCF in addition to the BRIEF provides useful information about a child’s cognitive functioning. Mean self-reported PedsPCF scores did not follow the different levels of school functioning. Interestingly, children attending special education scored the highest compared to the children with the other school functioning levels. This tendency for children with learning disabilities to overrate their cognitive abilities is a phenomenon that has been named ‘positive illusionary bias’ [40].

Strengths of our study include the large sample size in view of the rare diagnoses, the structured follow-up, the analysis of both proxy- and self-reported PedsPCF, and the inclusion of multiple neuropsychological tests. Our study not only has clinical implications but may also be instrumental to achieving international multicenter collaborations for the cross-cultural validation of standardized instruments [41].

Some limitations should be discussed. Firstly, some data from the questionnaires and the NPA were missing mainly due to organizational reasons and lack of time, seeing that the children had several assessments on a single day. Although the missing data might have caused an underestimation of the mean neuropsychological test scores and may have affected the associations between the PedsPCF and NPA scores, we assume that the relatively small number of missing data will not have biased the conclusions. Secondly, due to the exploratory nature of this study, we did not correct for multiple testing, which could have resulted in a type 1 error. However, even when correcting for multiple testing (with alpha/10 = 0.005), the majority of the results remain significant, preserving our conclusion. Furthermore, the school functioning category ‘regular with help’ is a very broadly defined concept that is probably more applicable to children in primary education than in secondary education in the Netherlands. To investigate the relation between school functioning level and the PedsPCF, other measurements, such as school test scores or teacher opinions, are necessary. Finally, the extraction of 10 items from the 30-item version of the PedsPCF to create a 10-item version may have resulted in an overestimation of the correlation coefficient.

In summary, we conclude that the proxy-reported PedsPCF reveals perceived cognitive problems for children who survived neonatal critical illness. In our population, the proxy-reported PedsPCF detected different and more cognitive problems than the proxy-reported BRIEF. Since there also appears to be a moderate correlation between the proxy-reported PedsPCF and the BRIEF, one might consider administering the PedsPCF instead of the BRIEF during follow-up, unless a more complete mapping of the perceived cognitive abilities is preferred. The proxy-reported PedsPCF can be used internationally to provide a first screening of these children’s cognitive functioning. With regard to clinical decision-making, the total proxy-reported PedsPCF scores can support the indication for extensive neuropsychological testing—and, therefore, could be a cost-effective implication. In the future, the short version of the PedsPCF may well become standard.

The self-reported PedsPCF proved not to be associated with outcomes from the NPA but can give complementary information about the cognitive problems these children experience in everyday life. We propose to implement a roadmap in which children scoring lower than or equal to −1 SD on the proxy-reported PedsPCF are referred for more detailed assessments (Appendix A: Flowchart). Considering the results of our study, one-third of the children would have been referred. For children whose parents report total PedsPCF scores higher than −1 SD, NPA may be considered after evaluation of anamnestic information from parents and teachers. Future studies should focus on exploring which specific anamnestic information is most informative to establish the need for detailed NPA, but should also focus on the cost-effectiveness of this approach.

## Figures and Tables

**Figure 1 children-09-00900-f001:**
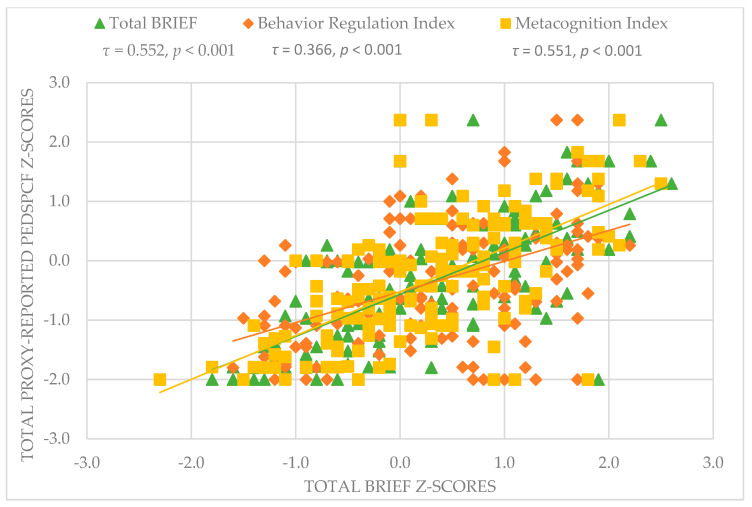
Association between proxy-reported BRIEF and proxy-reported PedsPCF scores. PedsPCF = pediatric perceived cognitive functioning, BRIEF = behavior rating inventory of executive function.

**Figure 2 children-09-00900-f002:**
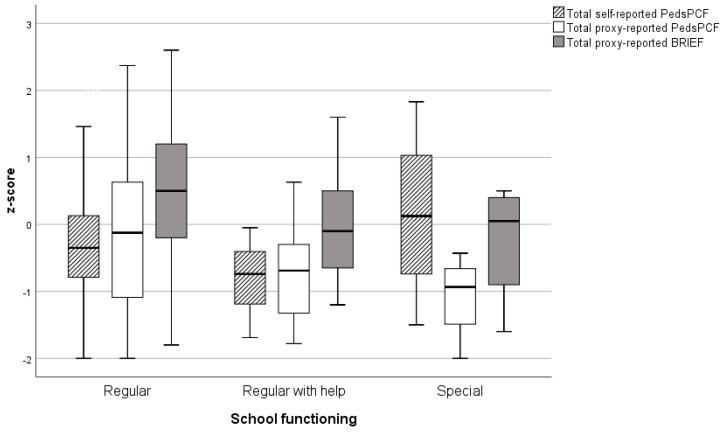
Total PedsPCF and BRIEF z-scores per level of school functioning. PedsPCF = pediatric perceived cognitive functioning, BRIEF = behavior rating inventory of executive function. *Y*-axis: levels of school functioning. Box-whisker plots represent the different questionnaires with the median, IQR, maximum and minimum scores (three positive outliers with total self-reported PedsPCF scores for children attending regular education are not shown).

**Table 1 children-09-00900-t001:** Demographic and clinical characteristics of 168 participants.

**Demographic Characteristics**	
Sex	
Male	99 (59%)
Female	69 (41%)
**Clinical characteristics**	
Diagnosis	
CDH	57 (34%)
without ECMO treatment	46 (81%)
with ECMO treatment	11 (19%)
EA	54 (32%)
CLM	20 (12%)
Neonatal ECMO, non-CDH	37 (22%)
Gestational age (weeks)	39.0 (37.1–40.4)
Birth weight (grams)	3150 (2725–3530)
Duration of initial hospital stay (days)	22 (13–48)
Duration of initial ventilation (days)	7 (2–16)
**At follow-up**	
Age at assessment	
8 years	57 (34%)
12 years	64 (38%)
17 years	47 (28%)
School functioning	
Regular	130 (78%)
Regular with help	22 (13%)
Special educational needs	14 (8%)
Missing	2 (1%)

Data are presented as number (%) or median (IQR). CDH = congenital diaphragmatic hernia, ECMO = extracorporeal membrane oxygenation, EA = esophageal atresia, CLM = congenital lung malformations.

**Table 2 children-09-00900-t002:** Self- and proxy-reported scores on perceived cognitive functioning and proxy-reported scores on executive functioning compared to population means.

	Measurement	Mean (*SD*)	*p*-Value	Effect Size (d/r)
Self-reported	PedsPCF 30-item	−0.35 (0.88)	<0.001	0.40 ^1^
Proxy-reported	PedsPCF 30-item	−0.36 (1.06)	<0.001	0.33 ^1^
	Total BRIEF	0.33 (0.98)	<0.001	0.34 ^2^
	BRI	0.37 (0.97)	<0.001	0.35 ^1^
	MCI	0.26 (0.96)	0.001	0.27 ^2^

Data are presented as mean z-scores (SD) (mean = 0, SD = 1). PedsPCF = pediatric perceived cognitive functioning, BRIEF = behavior rating inventory of executive function, BRI = Behavior Regulation Index, MCI = Metacognition Index. ^1^ One sample Wilcoxon signed rank test. Effect size = r. ^2^ One sample *t*-test. Effect size = Cohen’s d.

**Table 3 children-09-00900-t003:** Norm data comparison neuropsychological assessments.

	Variable	Mean (*SD*)	*p*-Value	Effect Size (d/r)
Intelligence	Total IQ	100 (18)	0.951	0.00 ^2^
Attention	TMTA t	0.30 (1.15)	<0.001 ***	0.37 ^1^
	Stroop IF	−0.23 (1.19)	0.051	0.16 ^1^
	DCT-ST	−0.93 (1.28)	<0.001 ***	0.60 ^1^
	DCT-SD	−1.17 (1.52)	<0.001 ***	0.60 ^1^
Verbal Memory	Total Digit Span	0.04 (1.00)	0.565	0.04 ^1^
	RAVLT total	−0.35 (1.26)	<0.001 ***	0.28 ^2^
	RAVLT recall	−0.64 (1.38)	<0.001 ***	0.40 ^1^
Visuospatial Memory	WNV-SS total	−0.21 (0.92)	0.004 **	0.23 ^2^
	WNV-SS forward	−0.07 (1.04)	0.298	0.08 ^1^
	RCFT immediate	−0.62 (1.23)	<0.001 ***	0.42 ^1^
	RCFT delayed	−0.76 (1.26)	<0.001 ***	0.51 ^1^
	RCFT recognition	−0.08 (1.21)	0.933	0.01 ^1^
Executive Functioning	BADS-C Key Search	0.18 (1.10)	0.052	0.15 ^1^
	BADS-C Modified Six Elements	−0.48 (0.81)	<0.001 ***	0.50 ^1^
	TMTB t	0.09 (1.08)	0.013 *	0.20 ^1^
	WNV-SS backwards	−0.29 (0.86)	<0.001 ***	0.35 ^1^
	PSI	−0.04 (1.13)	0.620	0.04 ^2^

Most data are presented as mean z-scores (*SD*) (mean = 0, *SD* = 1). Total IQ is presented as standard scores (mean = 100, *SD* = 15). IQ = Intelligence Quotient, TMTA t = Trail Making Test Section A time, Stroop IF = Stroop interference factor, DCT-ST = Dot Cancellation Test series time, DCT-SD = Dot Cancellation Test standard deviation series time, RAVLT: Rey Auditory Verbal Learning Test, WNV-SS: Wechsler Nonverbal Scale of Ability-Spatial Span, RCFT: Rey Complex Figure Test, BADS-C = Behavioral Assessment of the Dysexecutive Syndrome for Children, TMTB t: Trail Making Test Section B time, PSI: Processing Speed Index. ^1^ One sample Wilcoxon signed rank test. Effect size = *r*. ^2^ One sample *t*-test. Effect size = Cohen’s d. * *p* < 0.05, ** *p* < 0.01, *** *p* < 0.001.

**Table 4 children-09-00900-t004:** Association between proxy-reported PedsPCF score and neuropsychological assessments.

	Variable	Cases (n)	B (*SE*)	95% CI of B	*R* ^2^	*p*-Value
Intelligence	WISC-III-NL Total IQ	158	0.022 (0.004)	0.013–0.031	0.141	<0.001 ***
Attention	TMTA t	155	0.223 (0.071)	0.082–0.364	0.060	0.002 **
	Stroop IF	149	0.067 (0.074)	−0.079–0.213	0.006	0.365
	DCT-ST	151	0.139 (0.066)	0.009–0.269	0.029	0.036 *
	DCT-SD	151	0.106 (0.056)	−0.004–0.271	0.024	0.059
Verbal Memory	WISC-III-NL Total Digit Span	157	0.255 (0.083)	0.090–0.419	0.057	0.003 **
	RAVLT total	158	0.231 (0.065)	0.103–0.359	0.075	<0.001 ***
	RAVLT recall	158	0.188 (0.060)	0.070–0.305	0.060	0.002 **
Visuospatial Memory	WNV-SS total	155	0.266 (0.091)	0.085–0.446	0.052	0.004 **
	WNV-SS forwards	155	0.159 (0.082)	−0.003–0.322	0.024	0.055
	RCFT immediate	155	0.191 (0.067)	0.059–0.323	0.051	0.005 **
	RCFT delayed	155	0.191 (0.066)	0.061–0.321	0.052	0.004 **
	RCFT recognition	156	0.164 (0.071)	0.024–0.304	0.034	0.022 *
Executive Functioning	BADS-C Key Search	155	0.208 (0.076)	0.058–0.358	0.047	0.007 **
	BADS-C Modified Six Elements	88 ****	0.083 (0.144)	−0.202–0.369	0.004	0.563
	TMTB t	147	0.236 (0.083)	0.071–0.400	0.052	0.005 **
	WNV-SS backwards	155	0.275 (0.097)	0.082–0.467	0.049	0.005 **
	Processing Speed Index	158	0.319 (0.069)	0.183–0.456	0.120	<0.001 ***

B (*SE*) = Unstandardized coefficient β (standard error); 95% CI of B = 95% confidence interval of unstandardized β coefficient. PedsPCF = pediatric perceived cognitive functioning, IQ = Intelligence Quotient, TMTA t = Trail Making Test Section A time, Stroop IF = Stroop interference factor, DCT-ST = Dot Cancellation Test series time, DCT-SD = Dot Cancellation Test standard deviation series time, RAVLT = Rey Auditory Verbal Learning Test, WNV-SS = Wechsler Nonverbal Scale of Ability- Spatial Span, RCFT = Rey Complex Figure Test, BADS-C = Behavioral Assessment of the Dysexecutive Syndrome for Children, TMTB t = Trail Making Test Section B time. * *p* < 0.05, ** *p* < 0.01, *** *p* < 0.001, ****: Last subtest of the assessments, which was often not administered due to lack of time.

## Data Availability

Data presented in this study are available on reasonable request from the corresponding author.

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
