# Peer review of "Parent-Reported Perceived Cognitive Functioning Identifies Cognitive Problems in Children Who Survived Neonatal Critical Illness"

_children, 2022, doi:10.3390/children9060900_

Round 1

Reviewer 1 Report

Overall, I think the paper addresses an interesting topic. I have a few suggestions and requests for the authors to consider:

Introduction

- the association between cognitive difficulties and congenital anatomical anomaly was justified as an empirical relationship, but no discussion on the potential theoretical association between cognitive difficulties and this diagnosis was provided. I believe it is important to discuss it from a theoretical standpoint. In other words, does an hypothesis exist on why some of these children end up developing cognitive difficulties? Do these congenital anomalies occur with other risk factors of cognitive deficits?

Methods – participants

- It may be useful to report the inclusion criteria of the longitudinal follow-up program, in order to deepen the clinical picture of these patients. This may be of interest for clinicians working with this class of diagnoses.

Methods – measures

- Please provide more information on both PedsPCF and BRIEF items/scales. What area of functioning do they assess?

- Please provide more information on how the categories ‘regular with help’, and especially ‘special education’ were defined, as these definitions may imply different interventions in different countries.

Results

- It is not very clear to me whether age is a between-subject or a within-subject factor, as the sample is part of a longitudinal program. However, after reading Table 1, I am more inclined to say it’s a between-subject factor. In this case, it would be interesting to examine the associations of PedsPCF with BRIEF, outcomes of NPA, and school functioning also according to age category, being the range from 8 to 17 years old quite broad.

- I suggest reporting the formula of Wilcoxon and Cohen’s effect size in the data analysis paragraph, not in the table caption.

- Why are the associations between BRIEF and NPA measures not reported?

- Figure A1 is missing from the main text. It's not clear to me whether it’s supposed to appear only in the supplementary file.

Author Response

Dear Reviewer

We are grateful that you have taken the time to review our work. We believe that the manuscript has been improved by the changes we have now made. We hope you can agree with our considerations and adjustments. Please see the attachment for the revised version of the text. 

Introduction

- the association between cognitive difficulties and congenital anatomical anomaly was justified as an empirical relationship, but no discussion on the potential theoretical association between cognitive difficulties and this diagnosis was provided. I believe it is important to discuss it from a theoretical standpoint. In other words, does an hypothesis exist on why some of these children end up developing cognitive difficulties? Do these congenital anomalies occur with other risk factors of cognitive deficits?

Thank you for the comment regarding the potential theoretical association between cognitive difficulties and the discussed diagnoses. We added a phrase, see lines 42-45 which gives a possible explanation regarding the physiological pathway and underlying causes, which are thought to be the underlying mechanism, irrespective of diagnosis.

Methods – participants

- It may be useful to report the inclusion criteria of the longitudinal follow-up program, in order to deepen the clinical picture of these patients. This may be of interest for clinicians working with this class of diagnoses.

As mentioned in the introduction the only inclusion criteria for children joining the follow-up program are that they have major congenital anomalies, with the main group existing of children with esophageal atresia and congenital diaphragmatic hernia and/or having had intensive treatment at our IC-department (such as ECMO-treatment). The exclusion criteria for inclusion in this study are mentioned in lines 85-87.

Methods – measures

- Please provide more information on both PedsPCF and BRIEF items/scales. What area of functioning do they assess?

To provide more information regarding the PedsPCF and BRIEF we added some of the domains they question. In the added text, we also describe that the questions in the PedsPCF are unidimensional and we further explain the subscales of the BRIEF, see lines 105-107, 112-115.

- Please provide more information on how the categories ‘regular with help’, and especially ‘special education’ were defined, as these definitions may imply different interventions in different countries.

We thank the reviewer for this suggestion and added a specification of this classification system in lines 120-122.

Results

- It is not very clear to me whether age is a between-subject or a within-subject factor, as the sample is part of a longitudinal program. However, after reading Table 1, I am more inclined to say it’s a between-subject factor. In this case, it would be interesting to examine the associations of PedsPCF with BRIEF, outcomes of NPA, and school functioning also according to age category, being the range from 8 to 17 years old quite broad.

Age indeed is a between-subject factor since our data in this study are cross-sectional. In the future, we would hope to gather and analyse longitudinal data from our follow-up patients. In the results, we added a one-way ANOVA to show the non-significant differences between the PedsPCF scores when looking at the age groups, see lines 191-193.

- I suggest reporting the formula of Wilcoxon and Cohen’s effect size in the data analysis paragraph, not in the table caption.

In the revision, we reported the formulas in the data analysis paragraph, see lines 159-162.

- Why are the associations between BRIEF and NPA measures not reported?

In this paper, our focus was to explore the properties of the PedsPCF in these clinical groups. We felt that an extension towards the associations between BRIEF and NPA would distract too much attention from our main question. However, if the reviewer feels otherwise, we can obviously perform this analysis.  

- Figure A1 is missing from the main text. It is not clear to me whether it’s supposed to appear only in the supplementary file.

Figure A1 is mentioned in the main text in line 261. We did not include this figure in the main text, since we believe the text clarifies enough on its own and we felt that this figure added little to the message about the abbreviated PedsPCF.

Reviewer 2 Report

This manuscript reports analyses of the relationship between self- and parent-reported measures of cognitive functioning and a parent-reported measure of executive functioning, on the one hand, and neuropsychological assessment, on the other, for 168 children with congenital anomalies who were assessed at age 8, age 12, or age 17.  Of interest was whether these measures of perceived cognitive function might have clinical utility for determining whether neuropsychological assessment is warranted or necessary.

The description of the project seems relatively straightforward, but there are various points that require clarification and gaps that need to be filled.  Proper evaluation of the manuscript and its contribution would be facilitated by the inclusion of this additional information.

1.     Various scores are compared to a reference or normal population or normative data (e.g., on line 130), but as far as I can tell, no information is provided about what that reference population is or when and how data were obtained from them.

2.     The manuscript reports data collection from children who were 8, 12, or 17 years of age.  My understanding is that these children were 8, 12, or 17 years of age at the time all data (PedsPCF and BRIEF and NPA) were collected, but seeing this clearly stated would be helpful.  Are there any important differences between the age groups of which readers should be aware?

3.     As far as I can tell, none of the analyses were conditioned on age.  It seems plausible that there would be variation over age in the effectiveness with which children report with the PedsPCF.  There is really insufficient information about the psychometric properties of the PedsPCF (i.e., its reliability and validity and particularly how these might vary over subpopulations of children).  The authors say that a Dutch short form of the PedsPCF was “validated” in 2019, but they don’t say what “validated” means or on whom it was validated.  (I assume that data for this study was collected prior to the development of the shortened version.)

4.     The authors say that all statistical tests were two-sided. (lines 150-151).  Why?

5.     The authors say that a significance level of .05 was used for all tests. The authors conducted many tests. What steps, if any, were taken to protect against Type 1 errors?

6.     Should there be a reference to Figure 1 in the first paragraph of Section 3.3 (lines 183-188)?

7.     Nothing seems to be said about Table 3 other than about the first line, intelligence, on lines 192-193.  Does it make sense for there to not be a difference from the norm group in intelligence when there is a difference for so many things that surely contribute to intelligence?

8.     Data availability: Is there a reason to not make some version of the data available in a public repository (e.g., the Open Science Framework)

9.     Finally, an additional proofreading would be worthwhile. There is an occasional missing or incorrect word in the text.  P-values should not be reported as 0 (as on line 243).

Author Response

Dear reviewer

We are grateful that you have taken the time to review our work. We believe that the manuscript has been improved by the changes we have now made. We hope you can agree with our considerations and adjustments. 

  1. Various scores are compared to a reference or normal population or normative data (e.g., on line 130), but as far as I can tell, no information is provided about what that reference population is or when and how data were obtained from them.

In the supplementary material, all tests of the NPA (appendix B) are described together with references of the original test manuals. In these manuals, more information can be found regarding norm population data. We always selected the most recent and largest population data, and if available, Dutch reference data was applied. To clarify this, we added a sentence: lines 117-118.

  1. The manuscript reports data collection from children who were 8, 12, or 17 years of age.  My understanding is that these children were 8, 12, or 17 years of age at the time all data (PedsPCF and BRIEF and NPA) were collected, but seeing this clearly stated would be helpful.  Are there any important differences between the age groups of which readers should be aware?

We thank the reviewer for this suggestion. To clarify the moment when certain tests were administered, we added a sentence, see lines 92-93. To see if there were differences between the scores of these age groups on the PedsPCF, we conducted a one- way ANOVA. In the Results-section, we added this: lines 191-193.

  1. As far as I can tell, none of the analyses were conditioned on age.  It seems plausible that there would be variation over age in the effectiveness with which children report with the PedsPCF.  There is really insufficient information about the psychometric properties of the PedsPCF (i.e., its reliability and validity and particularly how these might vary over subpopulations of children).  The authors say that a Dutch short form of the PedsPCF was “validated” in 2019, but they don’t say what “validated” means or on whom it was validated.  (I assume that data for this study was collected prior to the development of the shortened version.)

In our paper of 2019, the data (1434 parents of 7- to 18-year-olds and 1181 children aged 8–18 years) used to validate the long version is the same as the data used to construct the short version. For more information about validity, we would like to refer to the articles of Lai (2011, ref. 14: Lai JS, Butt Z, Zelko F, Cella D, Krull KR, Kieran MW, et al. Development of a parent-report cognitive function item bank using item response theory and exploration of its clinical utility in computerized adaptive testing. J Pediatr Psychol. 2011;36(7):766-79) and Marchal (ref. 16).

  1. The authors say that all statistical tests were two-sided. (lines 150-151).  Why?

Since we had no strong assumptions about the directions of the associations,  all statistical tests were performed two-tailed.

  1. The authors say that a significance level of .05 was used for all tests. The authors conducted many tests. What steps, if any, were taken to protect against Type 1 errors?

Thank you for noticing this limitation. We did not correct for multiple comparisons due to the exploratory nature of this study. To address this, we added lines 341-345 to the discussion. Important to mention is that even when correcting for multiple testing (with a conservative Bonferroni correction), most results remain significant, so our main conclusions remain valid.

  1. Should there be a reference to Figure 1 in the first paragraph of Section 3.3 (lines 183-188)?

Thank you for noticing, we added a reference, see line 206-207.

  1. Nothing seems to be said about Table 3 other than about the first line, intelligence, on lines 192-193.  Does it make sense for there to not be a difference from the norm group in intelligence when there is a difference for so many things that surely contribute to intelligence?

That is indeed an interesting question. However, intelligence is a very broad concept, mainly suitable to predict general school functioning. Since in this paper, we only used one score,  children can compensate for weaker scores with scores in areas where they score higher. Furthermore, these intelligence test scores are obtained using the WISC-III which mainly does not include a Working Memory Index and a Fluid reasoning index as is the case in the newer WISC-V (In our country, we did not validate the WISC-IV). For this reason, the WISC-III does not take reasoning, executive functions, and attention into account as well as the WISC-V does.

  1. Data availability: Is there a reason to not make some version of the data available in a public repository (e.g., the Open Science Framework)

We our bound to certain Dutch legislation and are not allowed to provide all data to online platforms yet. In the future, we hope to find ways to share our data legally. However, we do allow individual researchers to use our data on written request.

  1. Finally, an additional proofreading would be worthwhile. There is an occasional missing or incorrect word in the text.  P-values should not be reported as 0 (as on line 243).

We thank the reviewer for bringing this to our attention, we have corrected some errors.

Round 2

Reviewer 1 Report

I thank the authors for the revisions. They have addressed my concerns. I don't have any further comments.